# A Real-World Data Observational Analysis of the Impact of Liposomal Amphotericin B on Renal Function Using Machine Learning in Critically Ill Patients

**DOI:** 10.3390/antibiotics13080760

**Published:** 2024-08-12

**Authors:** Ignasi Sacanella, Erika Esteve-Pitarch, Jessica Guevara-Chaux, Julen Berrueta, Alejandro García-Martínez, Josep Gómez, Cecilia Casarino, Florencia Alés, Laura Canadell, Ignacio Martín-Loeches, Santiago Grau, Francisco Javier Candel, María Bodí, Alejandro Rodríguez

**Affiliations:** 1Department of Pharmacy, Hospital Universitari Joan XXIII, 43005 Tarragona, Spain; isacanella.hj23.ics@gencat.cat (I.S.); eepitarch.hj23.ics@gencat.cat (E.E.-P.); lcanadell.hj23.ics@gencat.cat (L.C.); 2Department of Critical Care, Hospital Universitari Joan XXIII, 43005 Tarragona, Spain; jessicachaux1106@gmail.com (J.G.-C.); julen.berrueta@estudiants.urv.cat (J.B.); alejgarcia.hj23.ics@gencat.cat (A.G.-M.); mafloales@gmail.com (F.A.); mbodi.hj23.ics@gencat.cat (M.B.); 3Postgrado Medicina Crítica y Cuidado Intensivo, Facultad de Medicina, Fundación Universitari Ciencias de la Salud, Distrito Especial, Cra. 54 No.67A-80, Bogotá 111221, Colombia; 4Tarragona Health Data Research Working Group (THeDaR), 43005 Tarragona, Spain; 5Technical Secretary’s Department, Hospital Universitari Joan XXIII, 43005 Tarragona, Spain; josep.goal@gmail.com; 6Department of Pharmacy, Hospital de Pediatría Garrahan, Buenos Aires C1245, Argentina; ceciliacasarinomdq@gmail.com; 7Internal Medicine Department, Hospital Dr. Alejandro Gutiérrez, Venado Tuerto S2600, Argentina; 8Department of Intensive Care Medicine, Multidisciplinary Intensive Care Research Organization (MICRO), St James’ Hospital, D08 NHY1 Dublin, Ireland; drmartinloeches@gmail.com; 9Department of Pharmacy, Hospital del Mar, 08003 Barcelona, Spain; sgrau@parcdesalutmar.cat; 10Department of Medicine, Pompeu Fabra University, 08003 Barcelona, Spain; 11Clinical Microbiology & Infectious Diseases Department, Hospital Clínico Universitario San Carlos, 28040 Madrid, Spain; fj.candel@gmail.com; 12San Carlos Hospital Health Research Institute (IdISSC & IML), 28040 Madrid, Spain; 13Faculty of Medicine, Pere Virgili Health Research Institute, Rovira i Virgili University, 43005 Tarragona, Spain; 14Centre for Biomedical Research Network Respiratory Diseases (CIBERES), 43005 Tarragona, Spain

**Keywords:** liposomal amphotericin B, acute kidney injury, machine learning, critical care, antifungal agents, random forest

## Abstract

Background: Liposomal amphotericin B (L-AmB) has become the mainstay of treatment for severe invasive fungal infections. However, the potential for renal toxicity must be considered. Aims: To evaluate the incidence of acute kidney injury (AKI) in critically ill patients receiving L-AmB for more than 48 h. Methods: Retrospective, observational, single-center study. Clinical, demographic and laboratory variables were obtained automatically from the electronic medical record. AKI incidence was analyzed in the entire population and in patients with a “low” or “high” risk of AKI based on their creatinine levels at the outset of the study. Factors associated with the development of AKI were studied using random forest models. Results: Finally, 67 patients with a median age of 61 (53–71) years, 67% male, a median SOFA of 4 (3–6.5) and a crude mortality of 34.3% were included. No variations in serum creatinine were observed during the observation period, except for a decrease in the high-risk subgroup. A total of 26.8% (total population), 25% (low risk) and 13% (high risk) of patients developed AKI. Norepinephrine, the SOFA score, furosemide (general model), potassium, C-reactive protein and procalcitonin (low-risk subgroup) were the variables identified by the random forest models as important contributing factors to the development of AKI other than L-AmB administration. Conclusions: The development of AKI is multifactorial and the administration of L-AmB appears to be safe in this group of patients.

## 1. Introduction

Amphotericin B (AmB) is a natural antifungal extracted from *Streptomyces nodosus* belonging to the macrolide family that interacts with the ergosterol of the fungal membrane. AmB is a broad-spectrum antifungal indicated for the treatment of severe invasive fungal infections caused by *Cryptococcus neoformans*, *Histoplasma* spp., *Blastomyces* spp., *Coccidiodes* spp., *Mucor* spp., *Aspergillus* spp. (except *A. terreus*) and *Candida* spp. (except *C. lusitaniae*) [1,2,3]. However, species such as *Scedosporium* spp. [4,5] and some *Fusarium* spp. [6] are not sensitive to L-AmB. Despite its excellent spectrum of action, its use has been limited by the high incidence of nephrotoxicity when administered in the form of deoxycholate sodium salt [7].

The use of liposomal AmB (L-AmB) represents a strategy to reduce renal damage and to increase the total effective dose administered. The administration of L-AmB maintains its mechanism of action and broad antifungal spectrum, and over the years, it has been shown to have a low risk of resistance with a better adverse effect profile [8]. However, in critically ill patients, multiple confounding factors (i.e., sepsis, shock, vasoactive drugs and other nephrotoxic drugs) make it difficult to assess the impact of L-AmB administration on renal function. For this reason, observational studies with real-world data that include the greatest number of confounding factors can provide valuable clinical information on a subject that is still not well elucidated.

The aim of the current study was to evaluate renal function in adults (≥18 years of age) receiving L-AmB for more than 48 h in the Intensive Care Unit (ICU) and to analyze the possible risk factors associated with renal impairment. Our hypothesis was that L-AmB administration does not significantly affect renal function and, therefore, we can consider its administration safe.

## 2. Materials and Methods

### 2.1. Study Design and Participants

We conducted a retrospective, observational, single-center study over 6 years (2018–2024), including critically ill patients (≥18 years of age) consecutively admitted to a 30-bed ICU. In the present analysis, we included all critically ill subjects who required the administration of L-AmB for more than 48 consecutive hours due to a proven or probable fungal infection. Patients requiring continuous or intermittent renal replacement techniques (CRRT) at the time of beginning L-AmB and patients with a history of chronic kidney disease with or without the need for hemodialysis were excluded [9].

### 2.2. Definitions or Classification

Patients who had a serum creatinine value < 1.0 mg/dL obtained on the day before (day 0) the start of L-AmB infusion (day 1) were considered to have normal renal function (normal reference laboratory value < 1.5 mg/dL). Patients who had a serum creatinine value above >1.0 mg/dL on day 0 were considered to be at a high risk of developing AKI and they were analyzed as a particular subgroup with increased susceptibility to develop severe kidney dysfunction.

#### 2.2.1. Variables

The variables studied are shown in Table 1. These variables were obtained automatically from the clinical information system (CIS, Centricity Critical Care^®^, GE, Berlin, Germany), through the development of ETL (extract, transform and load) system specifically created using SQL (version 16.0) and Python (version 3.12.4).

The CIS automatically records all data from devices connected to the patient every two minutes, including hemodynamic variables, hourly diuresis, clinical parameters and laboratory values, as well as information on the medication administered. In addition, healthcare professionals register all patient-related information throughout the ICU stay.

#### 2.2.2. Definitions

-Baseline serum creatinine value

Because patients received L-AmB at different times during their ICU stay, the baseline creatinine value was considered to be that obtained on the day before (day 0) the start of L-AmB infusion (day 1).

-High risk of developing-AKI subgroup patients

Patients with serum creatinine levels above 1.0 mg/dL on the day prior to the start of L-AmB infusion (day 0) were considered as a subgroup of patients at a high risk of developing AKI with the start of L-AmB treatment (day 1). Conversely, those with serum creatinine levels below 1.0 mg/dL on day 0 were considered as a subgroup of patients without AKI and at a low risk of developing AKI. Both subgroups were analyzed separately.

### 2.3. Follow-Up and Endpoints

-Post-treatment renal dysfunction criteria

The diagnosis of renal dysfunction associated with L-AmB administration was made following the Acute Kidney Injury Network (AKIN) criteria for the diagnosis of acute renal failure described in the international KDIGO guidelines [10] (Appendix A). For the diagnosis of AKI, the worst serum creatinine value 72 h after L-AmB infusion was considered (day 3). This time window was selected because it is considered to be the minimum time necessary for the development of renal dysfunction.

### 2.4. Statistical Analysis

Our analysis plan is based on the following five steps:

Step 1: The behavior of serum creatinine and urea levels as well as urinary volume during the study period (day 1 to day 7) was analyzed. Categorical variables are presented as number and percentage (%), and continuous variables as median and interquartile range (Q1–Q3). To analyze differences between groups, the Mann–Whitney U-test (continuous) and Chi-square (dichotomous) test were used. Temporal differences between means were determined by Analysis of Variance (ANOVA) and paired ANOVA with Bonferroni correction.

Step 2: The linear association between variables of interest (creatinine, urea and total L-AmB dose administered) was determined by obtaining the Spearman (Rho) correlation coefficient due to the non-parametric distribution of the data.

Step 3: The incidence of AKI on day 3 of L-AmB administration was determined in the general population according to the definition considered. A bivariate comparison was made between the groups with and without AKI on day 3 of observation.

Step 4: The impact of the different variables on the development of AKI was established by multivariate analysis (multiple logistic regression). The result is shown as an odds ratio (OR) with a 95% confidence interval (95% CI). Values below 0.05 were considered significant. The assessment of model fit was performed using Akaike’s information criterion (AIC).

Due to the instability of linear regression models, a non-linear random forest (RF) model was developed to study the impact of covariates on the development of AKI [11]. The random forest algorithm is a powerful non-linear tree-based learning technique in machine learning. The performance of the RF model was evaluated by the out-of-bag (OOB) error. This method allows measuring the prediction error of random forests, boosted decision trees and other machine learning models using bootstrap aggregation.

Step 5: The analysis was performed similarly in the subgroup of patients with “low risk of AKI” on day 0, and in the subgroup of patients with “high risk of AKI”, consisting patients with a baseline (day 0) serum creatinine between 1.1 and 1.5 mg/dL.

All statistical analysis was performed using the free software R (version 4.3.0).

## 3. Results

### 3.1. Whole Population

A total of 84 patients were eligible, 17 (20.2%) of whom received L-AmB for less than 48 h and were therefore excluded. Finally, 67 patients were included in the present analysis (Figure 1). The admission diagnoses of the patients according to the CIE-10 coding can be seen in Appendix A. The median age was 61 years (53–71), including 67% male with a median severity level according to the SOFA score of 4 points. The median ICU stay was 17 days, with a crude mortality of 34.3%. The median dose of L-AmB received was 3 mg/kg/day with a median total dose of 1.2 g per day 7 (Table 1). The median infusion time was 2 h (1–3), and serum drug levels were not measured. During the observation period, there were no significant increases in serum creatinine, urea levels, calculated glomerular filtrate or daily urine output (Appendix A). Only a significant increase in serum urea values at day 3 and 4 from baseline was observed (Appendix A). It should be noted that the serum urea values were above the normal reference value at baseline (Table 1).

A weak positive correlation was observed between the serum creatinine and urea values at the baseline (Rho = 0.43, *p* = 0.002) and on the third day of L-AmB infusion (Rho = 0.45, *p* < 0.001) (Appendix A). No correlation was observed between the total L-AmB dose and serum creatinine or urea levels at day 3 (Appendix A).

### 3.2. Development of Renal Dysfunction

Fifty-six (83.6%) of the initial sixty-seven patients were still admitted to the ICU on the third day. Of these patients, 15 (26.8%) met the AKI criteria with a median serum creatinine value of 0.55 mg/dL (0.46–0.90) and a maximum value of 1.8 mg/dL. The characteristics of the patients on day 3 of ICU admission are shown in Table 2. No significant differences were observed between the groups.

### 3.3. Factors Associated with AKI Development

The association between AKI development and confounding variables was tested by developing multiple binary logistic regression models. The model was constructed with AKI development at day 3 as the dependent variable. Since no significant differences were observed between patients with and without AKI at day 3 (Table 2), the independent variables were those considered clinically relevant and clinically plausible. The linear models were very unstable with no significance and very wide confidence intervals. The best model obtained (AIC 28.4) was the one that included as independent variables the total L-AmB dose, SOFA score, age, sex, weight on admission, serum C-reactive protein and potassium level, all at day 3 of treatment (Appendix A). A more extensive explanation of the linear models can be found in Appendix A.

Due to the instability of linear models, a random forest model was developed to assess the impact of confounding variables on the development of AKI. Models developed with the inclusion of the variable total L-AmB dose at day 3 (Model 1; OOB = 31.3%) or total duration of L-AmB treatment (Model 2; OOB = 28.3%) showed that norepinephrine administration, severity measured by the SOFA score, furosemide, serum sodium and potassium concentrations were the variables with the highest contribution to the AKI development model (Figure 2). The total L-AmB dose at day 3 (model 1) and duration of L-AmB treatment (model 2) were the fifth- and seventh-highest contributing variables, respectively.

### 3.4. Subgroups Analysis: High and Low Risk of AKI

Of the 67 patients included, 15 (21.7%) had a baseline (day 0) serum creatinine level above 1.0 mg/dL. This subgroup was considered at a high risk for developing AKI. By contrast, the remaining 52 patients had a baseline (day 0) serum creatinine level ≤ 1.0 mg/dL, and these patients constituted the “low risk” group for AKI. The general characteristics of patients are shown in Table 1.

High risk of AKI subjects had a higher severity level as assessed by the SOFA score and as expected higher median serum creatinine and serum urea values, with no differences in urine output. By contrast, those patients had a lower calculated glomerular filtrate, body weight and fewer days of ICU stay and therefore received lower doses of L-AmB. However, although mortality was higher than in patients with a low risk of AKI, it did not reach statistical significance.

### 3.5. AKI Development in Patients with Low Risk of AKI at Baseline

No significant variations in serum creatinine levels, calculated glomerular filtrate and urine output were observed throughout the observation period (Appendix A). Although no significant variations in serum urea values were observed when considered overall, paired analysis showed significant increases on days 2, 3 and 4 compared to the baseline (Appendix A). Thirteen (25.0%) of the fifty-two patients met the AKI criteria (Table 3). No significant differences were observed between the groups, except in severity as measured by the SOFA score, which was slightly higher in the AKI group. The median serum creatinine values in the AKI and non-AKI groups were within the normal range. Only one patient met AKI II criteria who recorded a creatinine value at day 3 of 1.22 mg/dL (within the value considered normal).

### 3.6. Factors Associated with AKI Development in Patients with Low Risk of AKI

The random forest models implemented using the variable total dose or total duration of L-AmB showed an OOB estimate of the error rates of 30% and 25% respectively. In model 1, serum potassium levels, procalcitonin (PCT), total L-AmB dose at day 3 and albumin were the variables with the highest contribution to the model. In model 2, serum potassium levels, PCT, bilirubin and body weight were the major contributing variables. Interestingly, the total L-AmB duration was not a variable recognized as being of interest in the model and was excluded (Figure 3).

### 3.7. AKI Development in Patients with High Risk of AKI

Interestingly, when studying the evolution of serum creatinine levels, a significant decrease was observed during the observation period (Appendix A). As expected, the calculated glomerular filtration rate also increased significantly during the 7 days of observation (Appendix A).

No significant variations in urea levels (Appendix A) or urine volume (Appendix A) were observed. Only 2 (13.3%) patients met the AKI criteria on day 3 with a median serum creatinine value of 2.2 mg/dL. Risk modeling could not be performed due to the small number of patients.

## 4. Discussion

Our results using real-world data suggest that the administration of L-AmB in critically ill patients has a mild and clinically insignificant deleterious effect on renal function.

Different AKI criteria have been defined to identify hospitalized groups of patients at an increased risk of mortality and/or need for renal replacement therapy. The RIFLE and AKIN criteria were specifically developed for average-sized adults. In 2012, the global organization KDIGO attempted to harmonize the AKI staging criteria due to the heterogeneity of definitions formerly used in medical literature [9,10].

Considering the aforementioned issues, we relied on the AKIN scale, which has been previously employed, particularly in research concerning renal recovery following the administration of L-AmB [12,13], thereby facilitating cross-comparisons between studies.

Several factors have been identified in the development of AKI. Takazono et al. identified five AKI-associated factors in their Japanese study: the prior treatment with ACE/ARBs inhibitors or carbapenems, concomitant administration of catecholamines (as a state of shock) or immunosuppressants and ≥3.52 mg/kg/day of L-AmB dosing. In addition, they found that hypokalemia (serum potassium < 3.5 mEq/L) before starting L-AmB therapy was associated with severe AKI (stage II-III) [13]. Our results are consistent with the harmful effect of concurrent treatment with catecholamines (specifically norepinephrine in our scenario) and the dosage of L-AmB in the development of AKI, although we did not establish a specific mg/kg/day cutoff. We have not studied the impact of hypokalemia on AKI severity; however, our random forest model indicates that serum potassium plays an important role in AKI development. Despite this, other factors such as the previous exposure to ACE/ARB inhibitors or concomitant immunosuppressants did not significantly impact our AKI-development model.

By contrast, data from Personett et al. suggest that the L-AmB dose did not influence the probability of renal recovery, as renal worsening was not predicted by the daily dose of L-AmB at the time of AKI. They found no relationship between various factors (male sex, high weight, concomitant use of cyclosporine, vancomycin and ACE inhibitors) and the likelihood of recovering from a nephrotoxic event secondary to L-AmB exposure [14]. In our setting, we did not assess the renal function recovery, but among our cohort, none of these factors influenced the development of AKI. A Spanish multicenter study concluded that L-AmB treatment in ICU patients with impaired renal function had a minimal impact on kidney function, as evidenced by serum creatinine values [15]. These findings reaffirm L-AmB as a viable treatment option for IFI in critically ill patients, irrespective of their renal function at the initiation of therapy. Consistent with these data, our results demonstrate a similar trend as hypothesized by Álvarez-Lerma [15].

Due to the observed inconsistency of the linear regression model, we developed a non-linear tree-based technique using a supervised machine learning tool (random forest) to assess the true impact of several covariates on AKI development. Although the RF model evidenced a significant association with the development of AKI as measured by creatinine increase, this rise had no clinical implications, as most patients maintained serum creatinine values within normal limits, requiring no therapeutic intervention. Additionally, in low-risk patients for developing AKI (identified on day 0), the administration of L-AmB did not lead to elevated creatinine levels. Our random forest model for low-risk AKI patients demonstrated that the L-AmB dose contributes to the development of AKI, although other covariates (serum potassium and procalcitonin) showed greater impacts, highlighting the multifactorial nature of AKI development in critically ill subjects. Interestingly, among patients categorized as at a high risk of developing AKI, serum creatinine values decreased significantly during the period of L-AmB administration. This observation probably reflects the impact of therapy on the resolution of sepsis and secondary organ dysfunction and should not be attributed to a direct protective action of L-AmB administration.

Multiple studies have evaluated the nephrotoxic risks associated with L-AmB treatment. Ullmann et al. found an overall nephrotoxicity of 28.6% in patients treated with L-AmB, while it was more than 65% if they received AmB deoxycholate [16]. Hachem et al. observed a lower incidence of nephrotoxicity with L-AmB in patients with hematological malignancies and probable invasive aspergillosis in both primary (2.8%) and salvage (5.9%) treatments [17]. Wingard et al. reported an incidence of nephrotoxicity in the L-AmB subgroup that was not different regardless of the dose received, with 29.4% for 3 mg/Kg/d and 25.9% for 5 mg/Kg/d [18]. Walsh et al. observed an incidence of AKI of 50% defined by a serum creatinine value ≥ 1.5 times higher than the baseline, while 32% had a value ≥ 2 times higher than the baseline [7]. Finally, Cornely et al. observed no benefit in terms of the clinical efficacy of L-AmB but greater nephrotoxicity when using 3 mg/Kg/d or 10 mg/Kg/d regimens [19].

In our study, we observed a somewhat lower incidence of AKI ranging from 13% to 26%. These differences may be explained by the different populations considered, as only 15% of our patients had a history of immunosuppression in contrast to the other studies [16,17,18,19].

There are several limitations to this study. First, the reduced sample size and the single-center setting may lead to non-representative results. Although our study involved a small population, by limiting the study to a single center we aimed to reduce variability and ensure a more homogeneous sample. This approach provides clearer insights into the effects being studied. In addition, it should be taken into consideration that there are few published studies on L-AmB addressing the occurrence of secondary nephrotoxicity.

Second, we only assessed the change in creatinine levels during the first week of treatment. Therefore, an impact of L-AmB on renal function beyond 7 days of observation cannot be ruled out. However, the median number of treatment days was 6, so the impact of prolonged treatment beyond 7 days should be limited.

Third, we were only able to calculate the glomerular filtration rate using the Cockcroft formula, since, being a real-life study, the glomerular filtration rate is not normally calculated in patients with creatinine values within normal limits. However, the discrepancy of the estimated glomerular filtration rate with respect to the real glomerular filtration rate is well known. Furthermore, there is a high correlation between the calculated glomerular filtration rate and the creatinine values used for the diagnosis of AKI in the present analysis.

Fourth, new biomarkers of kidney damage were not used due to the retrospective nature of the study. However, as a future “prospective investigation”, the new biomarkers could provide valuable information about patients who develop AKI and may be able to differentiate whether this dysfunction is pseudo-AKI or real.

Fifth, we have not been able to obtain the baseline creatinine of the disease-free patients. However, the aim of the study was to evaluate the impact of L-AmB on creatinine after 72 h of administration independently of the pre-administration creatinine value.

Sixth, we did not determine the plasma concentration of magnesium in a sufficient number of patients to show that results and alterations in this ion could indicate nephrotoxicity [20]. However, no significant alterations of other ions such as potassium and sodium were observed. It is therefore possible that magnesium has a similar behavior to potassium in our patients.

Seventh, although bilirubin levels were slightly elevated only in patients with AKI, we did not determine transaminase levels, so secondary hepatitis cannot be completely ruled out. However, AmB commonly causes mild to moderate serum aminotransferase elevations and can cause hyperbilirubinemia, but acute, clinically apparent drug-induced liver injury from AmB therapy is exceedingly rare [21].

Finally, these results correspond to the population analyzed in our center and cannot be transferred to other populations or other centers without prior validation.

As a strength and to the best of our knowledge, this is the first study to incorporate supervised machine learning tools to relate the factors involved in L-AmB nephrotoxicity. However, further research is needed to shed light on the causes of L-AmB associated with AKI.

## 5. Conclusions

Our results with real-world data suggest that L-AmB treatment in critically ill adults did not result in a significant increase in serum creatinine levels during the first week of treatment. The intensity of the underlying infectious/inflammatory process and the presence of hemodynamic instability appear to be factors that contribute more to the development of AKI than L-AmB administration. Although our results need to be confirmed by further studies, the administration of L-AmB in critically ill patients seems safe and should not be delayed if a severe fungal infection is suspected.

## Figures and Tables

**Figure 1 antibiotics-13-00760-f001:**
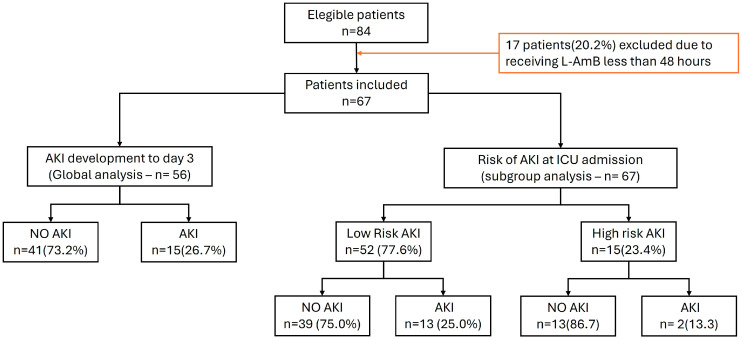
Flow chart of included patients.

**Figure 2 antibiotics-13-00760-f002:**
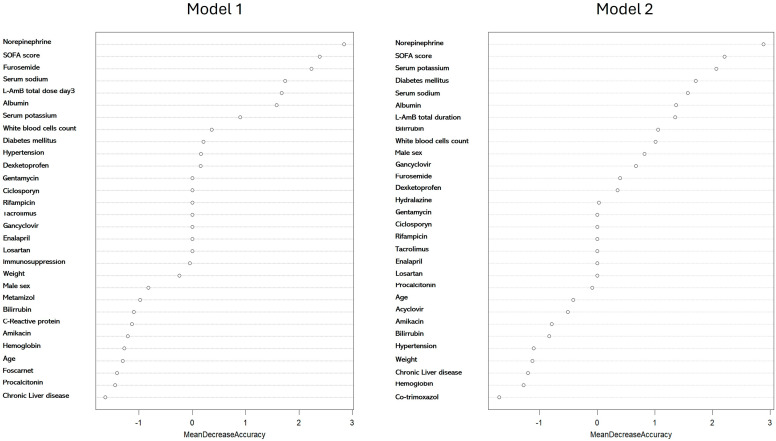
Contribution of each confounding variable according to the random forest (RF) model for variables associated with the development of AKI with the total dose of L-AmB administered at day 3 (Model 1) or the total duration of L-AmB administration (Model 2). This RF model was applied to the whole population of patients.

**Figure 3 antibiotics-13-00760-f003:**
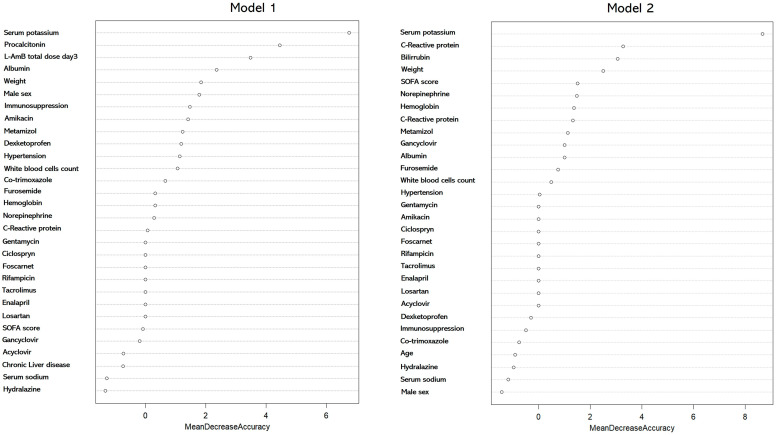
Contribution of each confounding variable according to the random forest (RF) model for variables associated with the development of AKI, considering either the total dose of L-AmB administered on day 3 (model 1) or the total duration of L-AmB administration (model 2). This RF model was applied to patients with a lower risk of AKI. Note: the total duration of L-AmB was not deemed important by the model and thus does not appear in the graph.

**Table 1 antibiotics-13-00760-t001:** Baseline characteristics of the 67 patients included in the analysis, categorized by the presence of acute kidney injury I (AKI I) at the start of the observation period (day 0).

Variable	Whole Population (*n* = 67)	Low Risk AKI (*n* = 52)	High Risk AKI (*n* = 15)	*p*-Value
**General**				
Age, median (Q1–Q3) years	61 (53–71)	61 (52–66)	65 (56–72)	0.25
Male sex, *n* (%)	45 (67.0)	33 (63.5)	12 (80.0)	0.35
Weight, median (Q1–Q3) kg	70 (64–80)	74 (67–80)	65 (53–72)	0.04
SOFA score, median (Q1–Q3)	4 (3–6.5)	5 (3–6)	7 (6–7)	0.01
Total diuresis in 24 h, median (Q1–Q3) mL	2100 (1500–2900)	2168 (1800–2880)	1825 (1300–3200)	0.49
**Laboratory**				
WBC count, median (Q1–Q3) ×10^3^	8.8 (2.0–15.5)	9.1 (7.1–15.4)	8.3 (7.1–15.9)	0.40
Lymphocytes count, median (Q1–Q3) ×10^3^	0.55 (0.23–1.2)	0.55 (0.23–1.24)	0.54 (0.42–0.88)	0.72
RCP, median (Q1–Q3) mg/dL	24 (13–30)	20 (13–29)	26 (24–30)	0.31
PCT, median (Q1–Q3) ng/mL	0.60 (0.25–2.05)	0.41 (0.22–1.59)	1.28 (0.53–12.4)	0.10
Serum Na^+^, median (Q1–Q3) mEq/L	138 (134–142)	139 (136–145)	137 (134–143)	0.29
Serum K^+^, median (Q1–Q3) mEq/L	3.5 (3.3–3.8)	3.5 (3.1–3.8)	3.7 (3.5–4.3)	0.17
Total bilirubin, median (Q1–Q3) mg/dL	0.6 (0.3–1.7)	0.5 (0.3–1.7)	0.7 (0.4–1.5)	0.69
Serum albumin, median (Q1–Q3) g/dL	2.7 (2.4–2.9)	2.7 (2.4–2.9)	2.7 (2.4–2.8)	0.90
Hemoglobin, median (Q1–Q3) g/dL	8.7 (7.8–10.0)	8.7 (7.7–9.1)	9.7 (8.1–10.8)	0.07
Serum Creatinine, median (Q1–Q3) mg/dL	0.61 (0.45–0.90)	0.54 (0.44–0.70)	1.43 (1.16–1.54)	0.001
Serum Urea, median (Q1–Q3) mg/dL	49 (33–79)	40 (30–65)	79 (65–104)	0.002
**Comorbidities**				
Diabetes mellitus, *n* (%)	13 (19.4)	9 (16.7)	4 (26.7)	0.45
Chronic liver disease, *n* (%)	7 (10.4)	5 (9.2)	2 (13.0)	0.64
Hypertension, *n* (%)	25 (37.3)	19 (35.2)	6 (40.0)	0.96
Immunosupression, *n* (%)	31 (46.3)	25 (46.3)	7 (46.7)	1.00
**Antifungal medication and concomitant drugs**				
L-AmB dose by Kg, median (Q1–Q3) mg/kg	3 (3–4)	3 (3–4)	3 (3.0–3.6)	0.71
L-AmB total dose at day 3, median (Q1–Q3) mg	672 (535–877)	675 (590–915)	552 (380–842)	0.12
L-AmB total dose at day 7, median (Q1–Q3) mg	1200 (710–1575)	1250 (808–1680)	855 (380–1420)	0.02
Norepinephrine, *n* (%)	9 (13.4)	5 (9.6)	4 (26.7)	0.10
Dexketoprofen, *n* (%)	17 (25.4)	15 (28.8)	2 (13.3)	0.32
Metamizole, *n* (%)	24 (35.8)	18 (34.6)	6 (40.0)	0.96
Furosemide, *n* (%)	28 (41.8)	23 (44.2)	5 (33.3)	0.63
Amikacin, *n* (%)	6 (8.9)	5 (9.6)	1 (6.7)	1.00
Acyclovir, *n* (%)	15 (22.4)	12 (22.2)	3 (20.0)	1.00
Foscarnet, *n* (%)	2 (3.0)	2 (3.8)	0 (0.0)	1.00
Co-trimoxazole, *n* (%)	14 (21.0)	13 (25.0)	1 (6.7)	0.16
Ganciclovir, *n* (%)	7 (10.4)	6 (11.5)	1 (6.7)	0.67
**Outcomes**				
ICU LOS, median (Q1–Q3) days	17.8 (8.4–37.3)	19.1 (8.9–42.9)	9.4 (7.7–20.1)	0.06
Days of L-AmB administration, median (Q1–Q3)	5.0 (3.1–8.2)	5.5 (3.7–8.0)	3.0 (2.0–7.0)	0.13
ICU crude mortality, *n* (%)	23 (34.3)	16 (30.8)	7 (46.7)	0.40

Abbreviations. ICU: intensive care unit; K^+^: potassium; L-AmB: liposomal amphotericin B; LOS: length of stay; PCT: procalcitonin; Q1: first quartile; Q3: third quartile; RCP: reactive C protein; Na^+^: sodium; WBC: white blood cell.

**Table 2 antibiotics-13-00760-t002:** Characteristics at day 3 of the ICU stay of the 56 patients who remained admitted to the ICU and received >72 h of L-AmB infusion according to the development of acute kidney injury I (AKI I) at the beginning of the observation period (day 0).

Variable	No AKI (*n* = 41)	AKI I (*n* = 15)	*p*-Value
**General**			
Age, median (Q1–Q3) years	61 (54–71)	61 (49–63)	0.39
Male sex, n (%)	27 (65.9)	10 (66.7)	1.00
Weight, median (Q1–Q3) kg	72 (65–80)	70 (70–82)	0.54
SOFA score at day 3, median (Q1–Q3)	5 (3–7)	6.5 (4–7)	0.62
Calculated glomerular filtrate, median (Q1–Q3) mL/min/m^2^	120 (87–171)	146(89–169)	0.59
Total diuresis at day 3, median (Q1–Q3) mL	2300 (1760–3380)	2150 (1630–2650)	0.52
**Laboratory**			
WBC count at day 3, median (Q1–Q3) ×10^3^	8.8 (4.7–15.6)	6.9 (2.5–14.0)	0.39
Lymphocytes count at day 3, median (Q1–Q3) ×10^3^	0.8 (0.4–1.2)	0.5 (0.3–0.7)	0.10
RCP at day 3, median (Q1–Q3) mg/dL	15 (6–25)	17 (10–24)	0.47
PCT at day 3, median (Q1–Q3) ng/mL	0.56 (0.31–1.30)	1.22 (0.58–13.5)	0.27
Serum Na^+^ at day 3, median (Q1–Q3) mEq/L	140 (136–144)	142 (139–146)	0.32
Serum K^+^ at day 3, median (Q1–Q3) mEq/L	3.7 (3.4–4.2)	3.4 (3.3–3.8)	0.14
Total bilirubin at day 3, median (Q1–Q3) mg/dL	0.7 (0.4–1.5)	1.1 (0.7–1.5)	0.37
Serum albumin at day 3, median (Q1–Q3)	2.6 (2.2–3.0)	2.8 (2.6–3.0)	0.64
Hemoglobin at day 3, median (Q1–Q3) g/L	8.5 (8.0–9.3)	8.5 (7.6–9.2)	0.96
Serum Creatinine at day 3, median (Q1–Q3) mg/dL	0.61 (0.54–0.83)	0.55 (0.46–0.90)	0.68
Serum Urea, median (Q1–Q3) mg/dL	69 (48–95)	66 (35–98)	0.72
**Comorbidities**			
Diabetes mellitus, *n* (%)	8 (19.5)	4 (26.7)	0.71
Chronic liver disease, *n* (%)	5 (12.2)	1 (6.6)	1.00
Hypertension, *n* (%)	15 (36.6)	7 (46.7)	0.70
Immunosuppression, *n* (%)	19 (46.3)	6 (40.0)	0.90
**Antifungal medication and concomitant drugs**			
L-AmB dose by kg, median (Q1–Q3) mg/kg	3.0 (3.0–3.5)	3.0 (3.0–5.0)	0.65
L-AmB total dose at day 3, median (Q1–Q3) mg	675 (600–828)	675 (458–1065)	0.75
Norepinephrine at day 3, *n* (%)	16 (39.0)	3 (20.0)	0.31
**Outcomes**			
ICU LOS, median (Q1–Q3) days	17.8 (8.8–32.5)	26.5 (12.4–54.0)	0.09
Days of L-AmB administration, median (Q1–Q3)	6 (4–9)	5 (2.5–7.5)	0.50
ICU crude mortality, *n* (%)	15 (36.6)	5 (33.3)	1.00

Abbreviations. ICU: intensive care unit; K^+^: potassium; L-AmB: liposomal amphotericin B; LOS: length of stay; PCT: procalcitonin; Q1: first quartile; Q3: third quartile; RCP: reactive C protein; Na^+^: sodium; WBC: white blood cell.

**Table 3 antibiotics-13-00760-t003:** Characteristics of the 52 patients with low risk of AKI at day 0 (baseline) according to the development of AKI I at 72 h of evolution in the ICU.

Variable	No AKI (*n* = 39)	AKI I (*n* = 13)	*p*-Value
**General**			
Age, median (Q1–Q3) years	61 (53–69)	59 (43–61)	0.22
Male sex, *n* (%)	24 (61.5)	9 (69.2)	0.74
Weight, median (Q1–Q3) kg	75.0 (63.8–80.0)	70.0 (70.0–75.2)	0.93
SOFA score at day 3, median (Q1–Q3)	4.0 (3.0–5.0)	5.0 (4.0–6.0)	0.04
Total diuresis in 24 h. at day 3, median (Q1–Q3) mL	2150 (1790–3150)	2290 (2040–2840)	0.76
**Laboratory**			
WBC count, median (Q1–Q3) ×10^3^	9.1 (5.5–14.2)	6.1 (1.3–13.7)	0.23
RCP, median (Q1–Q3) mg/dL	15.9 (12.2–19.4)	14.0 (11.2–24.3)	0.87
PCT, median (Q1–Q3) ng/mL	4.3 (1.9–5.3)	5.9 (1.2–7.5)	0.31
Serum Na^+^, median (Q1–Q3) mEq/L	139 (136–143)	142 (139.144)	0.26
Serum K^+^, median (Q1–Q3) mEq/L	3.7 (3.4–4.1)	3.4 (2.9–3.7)	0.07
Total bilirubin, median (Q1–Q3) mg/dL	1.5 (0.9–2.8)	2.0 (1.5–2.9)	0.24
Serum albumin, median (Q1–Q3)	2.7 (2.6–2.8)	2.6 (2.5–2.7)	0.63
Hemoglobin, median (Q1–Q3) g/L	8.5 (8.0–9.3)	8.5 (7.4–9.2)	0.39
Serum Creatinine, median (Q1–Q3) mg/dL	0.59 (0.48–0.67)	0.55 (0.50–0.87)	0.61
Serum Urea, median (Q1–Q3) mg/dL	59 (35–80)	66 (43–100)	0.44
**Comorbidities**			
Diabetes mellitus, *n* (%)	5 (12.8)	4 (30.8)	0.20
Chronic liver disease, *n* (%)	4 (10.3)	1 (7.7)	1.00
Hypertension, *n* (%)	15 (38.5)	4 (30.8)	0.74
Immunosuppression, *n* (%)	18 (46.2)	6 (46.2)	1.00
**Antifungal medication and concomitant drugs**			
L-AmB dose by Kg, median (Q1–Q3) mg/kg	3.0 (3.0–3.4)	3.0 (3.0–5.8)	0.11
L-AmB total dose at day 3, median (Q1–Q3) mg	675 (600–780)	675 (495–1350)	0.44
Norepinephrine, *n* (%)	12 (30.8)	2 (15.4)	0.47
**Outcomes**			
ICU LOS, median (Q1–Q3) days	17.8 (8.0–36.8)	26.5 (15.3–49.8)	0.09
Days of L-AmB administration, median (Q1–Q3)	6 (3.5–8.0)	5 (5.0–8.0)	0.70
ICU crude mortality, *n* (%)	12 (30.8)	4 (30.8)	1.00

Abbreviations. ICU: intensive care unit; K^+^: potassium; L-AmB: liposomal amphotericin B; LOS: length of stay; PCT: procalcitonin; Q1: first quartile; Q3: third quartile; RCP: reactive C protein; Na^+^: sodium; WBC: white blood cell.

## Data Availability

The data supporting the conclusions of this study are available from the Joan XXIII de Tarragona Hospital (Spain), but restrictions are placed on the free availability of these data by the health authorities of Catalonia, so they are not publicly available. However, the data can be obtained from the corresponding author (AR) upon reasonable request and with the permission of the Technical Secretary and the person responsible for data management at Joan XXIII de Tarragona Hospital (Spain).

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
