# Peer review of "A Real-World Data Observational Analysis of the Impact of Liposomal Amphotericin B on Renal Function Using Machine Learning in Critically Ill Patients"

_antibiotics, 2024, doi:10.3390/antibiotics13080760_

Round 1

Reviewer 1 Report

Comments and Suggestions for Authors

Sacanella et al. performed a retrospective study to investigate the effect of liposomal amphotericin B on the development of acute kidney injury (AKI) in critically ill patients by assessing serum creatinine levels and other factors. In their study, they utilized linear regression and non-linear machine learning approaches to identify contributing factors to the development of AKI in the sample. I find the study interesting. The manuscript is well-written, the methods section is well detailed, and the limitations are clearly stated at the end of the discussion section. Below are a few comments that the authors may need address:

The study lacks information regarding ethical approval. It is unclear whether the authors have obtained approval from an ethics committee. If approval was obtained, the reference number should be provided.

Title: change “critical” to “critically”. The authors may also consider declaring the observational study design they used in the title.

Abstract line 31: “Patients were analyzed globally and differentiated between patients with “low” and “high” risk of developing AKI with machine learning techniques”. I find the statement confusing as the authors reported in the methods section that the classification between “low” and “high” risk was based on the baseline serum creatinine level. A clarification is required.

I may suggest including factors that were identified to be associated with the development of AKI in the abstract and/or conclusion.

Comments on the Quality of English Language

I urge the authors to go through the manuscript and correct any grammatical of typographical errors.

Author Response

The study lacks information regarding ethical approval. It is unclear whether the authors have obtained approval from an ethics committee. If approval was obtained, the reference number should be provided.

REPLY: Thank you for your comment. The reviewer is correct, but ethical review and approval has been waived because this is an observational, retrospective study with automated collection of anonymized data for the purpose of assessing quality of care as part of the routine evaluation of the implementation of the PROA(antimicrobial optimization program) group measures. This information was added in the Informed Consent Statement section in the new manuscript.

Title: change “critical” to “critically”. The authors may also consider declaring the observational study design they used in the title.

Reply: Thank you for your comment. The title was changed following your suggestion

Abstract line 31: “Patients were analyzed globally and differentiated between patients with “low” and “high” risk of developing AKI with machine learning techniques”. I find the statement confusing as the authors reported in the methods section that the classification between “low” and “high” risk was based on the baseline serum creatinine level. A clarification is required.

REPLY: Thank you for the comment. The sentence was rewritten to clarify this concept as suggested. It can now read: AKI incidence was analyzed in the entire population and in patients with "low" or "high" risk of AKI based on their creatinine levels at the outset of the study, using Random Forest.

I may suggest including factors that were identified to be associated with the development of AKI in the abstract and/or conclusion.

REPLY: Thank you for your comment. We have rewritten the sentence following your suggestion. It is now readable: “Noradrenaline, SOFA score, and furosemide (general model) and potassium, CRP and PCT (low-risk subgroup) were the variables identified by the Random Forest models as important contributing factors to the development of AKI other than L-AmB administration”. Please note that the abstract only allows 200 words so the abstract has been modified globally to maintain this journal requirement.

Comments on the Quality of English Language

I urge the authors to go through the manuscript and correct any grammatical of typographical errors.

REPLY: Thank you for your comments and apologies for the typos. We have optimised the English of the manuscript

Reviewer 2 Report

Comments and Suggestions for Authors

The manuscript presents data on amphotericin B administration and the development of kidney injury. While not novel, it offers intriguing findings. To meet acceptance criteria for this journal, the following aspects should be addressed:

1.     LINE 28- “However, renal…” This sentence is unclear. I suggest rephrasing it for better clarity.

2.     LINE 33- This sentence is unclear. I suggest rephrasing it to include the specific name of the machine, clarify what "61 years" refers to (perhaps the median age), and add details about the participants such as the percentage of males and females and the median age for both genders.

3.     Please, I suggest to format the abstract as follows:

·      Background

·      Aims

·      Methods

·      Results

·      Conclusion

This format will help the reader to understand it better. 

4.     LINEs 47-50, chromoblastomycosis, Scedosporium spp, and some Fusarium spp, these are not sensitive to amphotericin B. I suggest adding these. 

5.     Please use the following references to incorporate information about the pathophysiology of amphotericin-induced nephrotoxicity into the introduction.. 

·      Sabra R, Branch RA. Amphotericin B nephrotoxicity. Drug Saf. 1990 Mar-Apr;5(2):94-108. doi: 10.2165/00002018-199005020-00003. PMID: 2182052.

·      https://doi.org/10.1155/2019/8629891

6.     In the results section, please provide a detailed explanation of how individuals were classified into low and high risk. I noticed this classification was mentioned in the materials and methods, and I recommend expanding on it for clarity.

7.     Table 1, please provide better quality.

8.     In the results section, please specify whether the levels of amphotericin were measured and, if so, include the relevant data.

9.     Please include the time of administration for amphotericin B (e.g., indicate the median time of administration was 3 hours).

10.  On line 93, please revise the first sentence to: "No patients met the criteria for AKI stage III."

11.  On line 94, please revise to: "No patients satisfied the criteria for AKI stage III."

12.  Table 2, please provide data if applicable about hypomagnesaemia and hypocalcemia.

13.  Please add to the results section that hepatitis from amphotericin B is rare. If applicable, include liver enzyme levels for patients with chronic liver disease. If no data is available, indicate "(data not shown)" and mention that hepatitis is infrequent with amphotericin B.

14.  On line 105, "2.2. Development of renal dysfunction" has been repeated. Should this be presented as another result?

15.  FIGURE 2 and 3, please provide better imaging quality. 

16.  LINE 184: In the "Discussion" section, the authors should begin with a paragraph summarizing the results. Furthermore, I strongly recommend comparing your results with those presented in the following manuscript: "Factor analysis of acute kidney injury in patients administered liposomal amphotericin B in a real-world clinical setting in Japan".

https://www.nature.com/articles/s41598-020-72135-y

17.  LINE 195, the reference [10] should be located at the end of the sentence, this applies for all the references in the manuscript.  

18.  The material and methods should be moved up before the results.

19.  Some words in this manuscript should be replaced with scientific and professional words. For instance, “met”, “we carried out”, etc. These should be replaced by more formal words, you can say “we conducted”

20.  LINE 292; “[6]”, should be at the end of the sentence.

21.  LINES 293-299, and 314-319, are both confusing, and looks like similar. Please rewrite those in a good way.

22.  I suggest for section 4 to be classified into:

1.     Study design and participants

2.     Definitions or classification

3.     Follow up and endpoints

4.     Statistical analysis 

23.  “Ethical consideration” has been mentioned at the end, no need to repeat it here. 

24.  Please, concise the statistical analysis.  

25.  LINES 327-333, for the classification of AKI grades, please create a table. 

26.  The conclusion is concise; please expand it by including more detail. Additionally, add a section titled "Prospective Research" to propose hypotheses regarding why some patients developed AKI, and whether this condition is pseudo-AKI or real. Consider suggesting the measurement of new markers such as CXCL9, NGAL, TGF-beta, or cystatin C based on your data.

27.  References should be expanded

Comments on the Quality of English Language

Needs minor editing, overall it's good. 

Author Response

The manuscript presents data on amphotericin B administration and the development of kidney injury. While not novel, it offers intriguing findings. To meet acceptance criteria for this journal, the following aspects should be addressed:

  1. LINE 28- “However, renal…” This sentence is unclear. I suggest rephrasing it for better clarity.

Reply: Thank you for your comment. We have rewritten the sentence to try to clarify the concept following your suggestion. It now reads: " The use of liposomal amphotericin B (L-AmB) for the treatment of severe invasive fungal infections has a potential risk of acute kidney injury(AKI)". Please note that the abstract only allows 200 words so the abstract has been modified globally to maintain this journal requirement.

  1. LINE 33- This sentence is unclear. I suggest rephrasing it to include the specific name of the machine, clarify what "61 years" refers to (perhaps the median age), and add details about the participants such as the percentage of males and females and the median age for both genders.

REPLY: Thank you very much for your comments. The reviewer is right, but unfortunately we are limited to using only 200 words in the unstructured abstract according to the journal rules. This makes it impossible for us to put more data in the abstract. We have indicated that the age is in median as well as the SOFA. We have rewritten the abstract again to follow the indications made by the different reviewers, but unfortunately we are unable to include more data, which are clearly presented in the tables of the manuscript. 

  1. Please, I suggest to format the abstract as follows:
  • Background
  • Aims
  • Methods
  • Results
  • Conclusion

This format will help the reader to understand it better.

REPLY: Thank you for your comments. The reviewer is right and we also believe that a structured abstract is more understandable for readers. However, the rules of the journal are clear in this respect and indicate to send an unstructured abstract of only 200 words.

  1. LINEs 47-50, chromoblastomycosis, Scedosporium spp, and some Fusarium spp, these are not sensitive to amphotericin B. I suggest adding these.

REPLY: Thank you for your very accurate comments. We have mentioned in one sentence "While species such as Scedosporium spp [ ], and some Fusarium spp[ ] are not sensitive to L-AmB" . However, we have not included Chromoblastomycosis because amphotericin B is not considered a first-line treatment due to its significant nephro- and cardiotoxicity, a major contraindication for long-term treatment,but it is effective in Chromoblastomycosis according to different publications.

FLAVIO QUEIROZ-TELLES et al. Chromoblastomycosis: an overview of clinical manifestations, diagnosis and treatment.  Medical Mycology February 2009, 47 (Special Issue), 3-15.

2.- George Kurien; Kavin Sugumar; Nishad C. Sathe; Veena Chandran. Chromoblastomycosis.  Treasure Island (FL): StatPearls Publishing; 2024 - Last updated: 1 March 2024.

3.- https://www.who.int/news-room/fact-sheets/detail/chromoblastomycosis

  1. Please use the following references to incorporate information about the pathophysiology of amphotericin-induced nephrotoxicity into the introduction.
  • Sabra R, Branch RA. Amphotericin B nephrotoxicity. Drug Saf. 1990 Mar-Apr;5(2):94-108. doi: 10.2165/00002018-199005020-00003. PMID: 2182052.
  • https://doi.org/10.1155/2019/8629891

REPLY: Thank you very much. The information and the references have been updated according to your suggestion.

  1. In the results section, please provide a detailed explanation of how individuals were classified into low and high risk. I noticed this classification was mentioned in the materials and methods, and I recommend expanding on it for clarity.

REPLY: Thank you for your comment. As requested by the reviewer we have rewritten the paragraph to clarify how patients have been classified into low and high risk of AKI. It now reads: "Of the 67 patients included, 15 (21.7%) had a baseline (day 0) serum creatinine level greater than 1.0 mg/dL. This subgroup was considered to be at 'high risk' of developing AKI. In contrast, the remaining 52 patients had a baseline (day 0) serum creatinine level < 1.0 mg/dL; these patients constituted the "low risk" group for AKI.

  1. Table 1, please provide better quality.

REPLY: Thank you for your comment. Tables 1, 2 and 3 have been attached in word format for better quality.

  1. In the results section, please specify whether the levels of amphotericin were measured and, if so, include the relevant data.

REPLY: Thank you for your comment. L-AmB levels were not determined in the patients. This has been added to the results as you suggested.

  1. Please include the time of administration for amphotericin B (e.g., indicate the median time of administration was 3 hours).

REPLY: Thank you for your comment. The average infusion time was 2 hours (1-3 hours). This information was aggregated in results  

  1. On line 93, please revise the first sentence to: "No patients met the criteria for AKI stage III."

REPLY: Thank you for your comments. This sentence was removed from the manuscript

  1. On line 94, please revise to: "No patients satisfied the criteria for AKI stage III."

REPLY: Thank you for your comments. This sentence was removed from the manuscript

  1. Table 2, please provide data if applicable about hypomagnesaemia and hypocalcemia.

REPLY: The reviewer is right in his observation about the importance of magnesium and calcium. However, these variables had more than 70% missing data in our real-life database and therefore no missing data have been imputed and both variables had to be removed from the final database and we cannot present their results. This has been added to the limitations section.” Sixth, we have not determined the plasma concentration of magnesium in a number necessary to show results and alterations of this ion could be an indicator of nephrotoxicity. However, no significant alterations of other ions such as potassium and sodium were observed. It is therefore possible that magnesium has a similar behaviour to postassium in our patients”.

  1. Please add to the results section that hepatitis from amphotericin B is rare. If applicable, include liver enzyme levels for patients with chronic liver disease. If no data is available, indicate "(data not shown)" and mention that hepatitis is infrequent with amphotericin B.

REPLY: Thank you for your comment and valuable input. However, we have no data on transaminase levels in our patients. We only present data on total Bilirubin in the tables. We believe it is appropriate to add this in limitations where it reflects that hepatitis secondary to AmB administration is an exceptional situation. The new limitation can be read as “Eighth, although bilirubin levels were slightly elevated only in patients with AKI. We did not determine transaminase levels so secondary hepatitis cannot be totally ruled out. However, AmB commonly causes mild to moderate serum aminotransferase elevations and can cause hyperbilirubinemia, but acute, clinically apparent drug induced liver injury from AmB therapy is exceedingly rare”

  1. On line 105, "2.2. Development of renal dysfunction" has been repeated. Should this be presented as another result?

REPLY: Thank you for your comment and apologies for this unintentional error. The duplication has been fixed.

  1. FIGURE 2 and 3, please provide better imaging quality.

REPLY: Thank you for your comment, the figures were attached in higher quality.

  1. LINE 184: In the "Discussion" section, the authors should begin with a paragraph summarizing the results. Furthermore, I strongly recommend comparing your results with those presented in the following manuscript: "Factor analysis of acute kidney injury in patients administered liposomal amphotericin B in a real-world clinical setting in Japan".

https://www.nature.com/articles/s41598-020-72135-y

REPLY: Thank you for your comment. We have rewritten the beginning of the discussion following your suggestion. Takazono's study is mentioned and compared in the third paragraph of the discussion (page 9, line 255).

  1. LINE 195, the reference [10] should be located at the end of the sentence, this applies for all the references in the manuscript.

REPLY: Thank you for your comment. The reference has been placed at the end of the sentence following your suggestion.

  1. The material and methods should be moved up before the results.

REPLY: Thank you for your comment. We agree with the reviewer. However, the rules of the journal require this layout of the sections.

  1. Some words in this manuscript should be replaced with scientific and professional words. For instance, “met”, “we carried out”, etc. These should be replaced by more formal words, you can say “we conducted”

REPLY: Thank you for your comments. The manuscript was improved with professional terms.

  1. LINE 292; “[6]”, should be at the end of the sentence.

REPLY: Thank you for your comment. The reference was placed at the end of the sentence

  1. LINES 293-299, and 314-319, are both confusing, and looks like similar. Please rewrite those in a good way.

REPLY: Thank you for your comments. The discussion was rewritten to improve readability.

  1. I suggest for section 4 to be classified into:
  2. Study design and participants
  3. Definitions or classification
  4. Follow up and endpoints
  5. Statistical analysis

REPLY:  Thank you for your comment. The section was grouped according to your suggestion

  1. “Ethical consideration” has been mentioned at the end, no need to repeat it here.

REPLY: Thank you for your comment. The paragraph on ethical considerations has been removed in this section.

  1. Please, concise the statistical analysis.

REPLY: Thank you for your comments. We have reduced the information in this section and added more information about the machine learning model in Appex A following your comment and also responding to reviewer 4's comment.

  1. LINES 327-333, for the classification of AKI grades, please create a table.

REPLY: Thank you for your comment. Following your suggestion we have made a table with the AKIN classification and added it to Appex A.

  1. The conclusion is concise; please expand it by including more detail. Additionally, add a section titled "Prospective Research" to propose hypotheses regarding why some patients developed AKI, and whether this condition is pseudo-AKI or real. Consider suggesting the measurement of new markers such as CXCL9, NGAL, TGF-beta, or cystatin C based on your data.

REPLY: Thank you for your comments. We have expanded the conclusion following your suggestion. In addition, we have added the suggestion of a future Prospective Research within the limitations.

  1. References should be expanded

REPLY: Thank you for your comment. The number of references was increased as suggested.

Reviewer 3 Report

Comments and Suggestions for Authors

This paper is single Center study related to evaluate the incidence of acute kidney injury (AKI) in critically ill patients receiving L-AmB for more than 48 hours. The authors included 67 patients and concluded that their results with real-world data suggest that L-AmB treatment in critically ill adults is safe and did not result in a significant increase in serum creatinine levels during the first week of treatment.

This study may be useful for physicians, but I have some comments and suggestions.

In Fig.1 in flow chart, one of the box High risk of AKI at ICU submission is branched to Low risk AKI and High risk AKI. It would be better to rename box High risk of AKI at ICU submission to Evaluated risk of AKI at ICU submission.

Table 1 contains baseline characteristics of patients including comorbidities, but there are no information about their primary diagnosis why they are admitted to ICU, they are treated only as critically ill patients.

Follow up period was 7 days, and the median number of treatment days was 6. The terminal half-life L-AmB in plasma is about 152 hours (Bekersky I et al. 2002), that is more than 6 days, so why did you choose follow up period to be only 7 days?

One limitation of the study, noted at the end of Discussion is the fact that GFR was not calculated because it is not the practice when creatinine levels are normal. But since renal function and AKI were assessed in this manuscript, it would be appropriate to at least calculate an estimated GFR.

Subsection 2.2 was repated twice (lines 92-97 are the same as 105-110).

Comments on the Quality of English Language

Minor editing of English language is required.

Author Response

Comments and Suggestions for Authors

This paper is single Center study related to evaluate the incidence of acute kidney injury (AKI) in critically ill patients receiving L-AmB for more than 48 hours. The authors included 67 patients and concluded that their results with real-world data suggest that L-AmB treatment in critically ill adults is safe and did not result in a significant increase in serum creatinine levels during the first week of treatment.

This study may be useful for physicians, but I have some comments and suggestions.

In Fig.1 in flow chart, one of the box High risk of AKI at ICU submission is branched to Low risk AKI and High risk AKI. It would be better to rename box High risk of AKI at ICU submission to Evaluated risk of AKI at ICU submission.

REPLY: Thank you for your comment. The reviewer is right and we apologise for the error. Now the branch of included patients is divided into Risk of AKI at ICU admission (subgroup analysis - n= 67) and these are subdivided into high and low risk. We believe this is now more understandable and clearer for the readers.

Table 1 contains baseline characteristics of patients including comorbidities, but there are no information about their primary diagnosis why they are admitted to ICU, they are treated only as critically ill patients.

REPLY: Thank you for your comment. The reviewer is right. We have added a table with the entry diagnoses in APPEX A.

Follow up period was 7 days, and the median number of treatment days was 6. The terminal half-life L-AmB in plasma is about 152 hours (Bekersky I et al. 2002), that is more than 6 days, so why did you choose follow up period to be only 7 days?

REPLY: Thank you for your important comment, The reviewer is right. There is no consensus on when to determine the impact of L-AmB on renal function. In view of this lack of evidence, we consider a 7-day observation period for the following reasons

1.- The number of patients receiving L-AmB for more than 7 days is low in our sample, a more distant observation would greatly diminish the power of the results.

2.- As most authors do not mention the observation time in their studies, there is no clear reference. For example Takazono et al. defined AKI as an increase of = 1.5 times in 7 days or = 0.3 mg/dL in 2 days in serum creatinine levels.  It is recognised that the nephrotoxic effect of L-AmB can be very early so we proposed to define the presence of AKI at day 3 where it is assumed that the drug has reached steady state. However, we monitored values up to 7 days to assess whether it was necessary to consider a point other than day 3.

3.- It is to be expected that changes in renal function that appear early with L-AmB administration may be more closely related than those that appear late, given the multifactorial aetiology of AKI in critically ill patients.

One limitation of the study, noted at the end of Discussion is the fact that GFR was not calculated because it is not the practice when creatinine levels are normal. But since renal function and AKI were assessed in this manuscript, it would be appropriate to at least calculate an estimated GFR.

REPLY: Thank you for your comment. Following your suggestion, the glomerular filtration rate was calculated using the cockcroft-gault formula. The results obtained are shown in the main tables of the manuscript and figures have been added in APPEX A.

Subsection 2.2 was repated twice (lines 92-97 are the same as 105-110).

REPLY: Thank you for your comment and apologies for this unintentional error. The duplication has been fixed

Reviewer 4 Report

Comments and Suggestions for Authors

The explanation of the machine learning methods and statistical analyses is not detailed enough. Please provide more information so readers can understand exactly how the models were trained, validated, and tested.

Comments on the Quality of English Language

The English language is fine overall, a minor editing of the manuscript may be required. 

Author Response

Comments and Suggestions for Authors

The explanation of the machine learning methods and statistical analyses is not detailed enough. Please provide more information so readers can understand exactly how the models were trained, validated, and tested.

REPLY: Thank you for your comment. Following the comments of reviewer 2 we have had to shorten and make the material and methods section more concise. Following your comment we have made an extensive description of the methodology of the analysis and modelling performed in APPEX A where readers can find all the information related to the study.  

Comments on the Quality of English Language

The English language is fine overall, a minor editing of the manuscript may be required.

REPLY: Thank you for your comment, the English was improved throughout the document.

Round 2

Reviewer 1 Report

Comments and Suggestions for Authors

I have no more comments.

Author Response

Dear Reviewer

Thanks for the help in improving the manuscript 

Reviewer 2 Report

Comments and Suggestions for Authors

Dear authors,

My objective is to enhance the manuscript to its fullest potential. Please adhere to the reviewers' suggestions seriously. You may expand the abstract to 250-300 words if needed; the editor will understand the reviewers' perspective.

Please review the comments we provided and make the necessary changes. The manuscript is not currently of publishable quality.

The editor will understand the reviewers' suggestions and their context. Some manuscripts may require exceptions to standard rules to improve their prospects. Please implement these changes to enhance the manuscript's quality.

In order to proceed with publication in the high-impact journal 'Antibiotics,' your manuscript needs to be improved.

Please ensure that all abbreviations in the abstract are expanded.

Sincerely,

Author Response

Dear Reviewer,
We extend our sincerest apologies if our responses have been unclear and have caused you distress.  We are confident that the comments were intended to enhance the manuscript. However, we would like to note that, based on our experience, the journal's guidelines are often quite rigorous. This is why we were unable to implement the requested changes. Once we had discussed this with the editor and received authorisation for the modifications, we edited the manuscript once more in accordance with your suggestions. We hope that it meets with your approval and can be published

Reviewer 3 Report

Comments and Suggestions for Authors

The authors responded to all comments.

Comments on the Quality of English Language

Minor editing of English language required.

Author Response

Dear Reviewer, Thank you for your help in improving the manuscript. We have edited the manuscript again to improve it.